# Urinary Metabolomic Profile in Children with Autism Spectrum Disorder

**DOI:** 10.3390/ijms26052254

**Published:** 2025-03-03

**Authors:** Joško Osredkar, Kristina Kumer, Uroš Godnov, Maja Jekovec Vrhovšek, Veronika Vidova, Elliott James Price, Tara Javornik, Gorazd Avguštin, Teja Fabjan

**Affiliations:** 1Institute of Clinical Chemistry and Biochemistry, University Medical Centre Ljubljana, 1000 Ljubljana, Slovenia; josko.osredkar@kclj.si (J.O.); kristina.kumer@kclj.si (K.K.); tara.javornik@gmail.com (T.J.); 2Faculty of Pharmacy, University of Ljubljana, 1000 Ljubljana, Slovenia; 3Faculty of Mathematics, Natural Sciences and Information Technologies, University of Ljubljana, 6000 Koper, Slovenia; uros.godnov@fm-kp.si; 4Center for Autism, Unit of Child Psychiatry, University Children’s Hospital, University Medical Centre Ljubljana, 1000 Ljubljana, Slovenia; maja.jekovec@kclj.si; 5RECETOX, Faculty of Science, Masaryk University, 61137 Brno, Czech Republic; veronika.vidova@recetox.muni.cz (V.V.); elliott.price@recetox.muni.cz (E.J.P.); 6Environmental Exposure Assessment Research Infrastructure-Czech Republic (EIRENE-CZ), 60200 Brno, Czech Republic; 7Department of Microbiology, Biotechnical Faculty, University of Ljubljana, 1230 Domžale, Slovenia; gorazd.avgustin@bf.uni-lj.si

**Keywords:** autism spectrum disorder, tryptophan, kynurenine, CARS

## Abstract

Autism spectrum disorder (ASD) has been associated with disruptions in tryptophan (TRP) metabolism, affecting the production of key neuroactive metabolites. Investigating these metabolic pathways could yield valuable biomarkers for ASD severity and progression. We included 44 children with ASD and 44 healthy children, members of the same family. The average age in the ASD group was 10.7 years, while the average age in the control group was 9.4 years. Urinary tryptophan metabolites were quantified via liquid chromatography—mass spectrometry operating multiple reaction monitoring (MRM). Urinary creatinine was analyzed on an Advia 2400 analyzer using the Jaffe reaction. Statistical comparisons were made between ASD subgroups based on CARS scores. Our findings indicate that children with ASD have higher TRP concentrations (19.94 vs. 16.91; *p* = 0.04) than their siblings. Kynurenine (KYN) was found at higher levels in children with ASD compared to children in the control group (82.34 vs. 71.20; *p* = 0.86), although this difference was not statistically significant. The ASD group showed trends of higher KYN/TRP ratios and altered TRP/ indole-3-acetic acid (IAA) and TRP/5-hydroxyindoleacetic acid (5-HIAA) ratios, correlating with symptom severity. Although the numbers of the two groups were different, our findings suggest that mild and severe illnesses involve separate mechanisms. However, further comprehensive studies are needed to validate these ratios as diagnostic tools for ASD.

## 1. Introduction

Autism spectrum disorder (ASD), also known as autism, is a neurodevelopmental disorder that affects behavior, speech, and social interaction. It is characterized by a broad spectrum of symptoms and degrees of impairment that vary significantly from person to person. While the precise etiology of ASD is still unknown, research suggests that it is likely caused by a complex interaction between genetic and environmental factors [1].

Recent research has pointed to tryptophan (TRP) metabolism as a key factor in the etiopathogenesis of ASD [2], however, high heterogeneity is observed across studies [3] and associations with ASD’s pathophysiology remain inconclusive [4].

TRP is an essential amino acid involved in the production of critical neuroactive compounds such as serotonin (5-HT), melatonin, and kynurenine (KYN), all of which play crucial roles in brain development and function [5].

The gut microbiota, gut–brain axis, nutrition, TRP metabolites, and biosynthetic gene polymorphisms are only a few of the intricate relationships that link TRP metabolism to ASD. Nevertheless, it is uncertain whether the association is causal, responsive, or a proxy, and the mechanism is unknown.

This study was carried out to fill important gaps in knowledge, as there have not been many thorough investigations on tryptophan metabolites in children with ASD. We sought to shed light on the metabolic changes connected to gut microbiota, immunological response, and sex-based disparities by examining metabolites and ratios such as KYN/TRP and TRP/5-HT. This study emphasizes the need for additional investigation to confirm these results, create normative data, and examine how tryptophan pathways relate to the symptomatology of ASD.

Disruptions in the gut microbiota, frequently seen in individuals with ASD, can significantly affect TRP metabolism [6]. These microbiota disruptions can alter the balance between 5-HT and KYN synthesis, with potential consequences for neurodevelopmental and behavioral traits [7,8]. Studies have shown that beneficial bacterial species, such as *Lactobacillus* and *Bifidobacterium*, which are important for maintaining gut health and regulating the gut–brain axis, are reduced in ASD [9]. This reduction leads to a decrease in the production of short-chain fatty acids and other metabolites essential for neurodevelopment. An imbalance in TRP metabolism, driven by microbiota disruptions, can shift the metabolic pathway from 5-HT production towards the kynurenine pathway, resulting in the accumulation of neuroactive metabolites. These metabolites contribute to neuroinflammation and neurotoxicity, both of which are critical components in the pathogenesis of ASD [10]. This feedback loop, where gut health affects neurodevelopment, further exacerbates metabolic abnormalities seen in ASD. Disruptions in the microbiota contribute to metabolic imbalances, which in turn affect neurodevelopmental outcomes, highlighting the intertwined relationship between the gut and brain in ASD.

Polymorphisms in gene-encoding enzymes, like tryptophan hydroxylase and indoleamine 2,3-dioxygenase (IDO), influence TRP metabolism. Variations in these enzymes can alter the rates of metabolite production. Inflammatory processes, particularly those involving pro-inflammatory cytokines, can activate IDO, leading to altered tryptophan breakdown and a shift towards kynurenine pathway metabolites, which are often linked to immune dysregulation in ASD [11]. Furthermore, stress and psychiatric conditions can modulate tryptophan metabolism by altering hormone levels and neurotransmitter activity, complicating the metabolic balance in ASD. Stress-induced activation of the hypothalamic–pituitary–adrenal (HPA) axis leads to elevated cortisol levels, which can shift tryptophan metabolism towards the kynurenine pathway, reducing 5-HT production. This mechanism has been associated with mood disorders and ASD [12]. There is currently no solid evidence from the comparisons between healthy (unrelated) groups and ASD, and there are numerous confounders.

Variations in TRP metabolites (such as KYN, 5-HT, melatonin, and indole derivatives) have been found between children with ASD and healthy controls [13,14]. Children with ASD have been reported to exhibit reduced levels of TRP and its metabolites in cord blood, which may contribute to the symptoms of repetitive behaviors and social deficits [3,4,15,16].

TRP is metabolized through three main pathways (Table 1; Figure 1):

Kynurenine Pathway: Enzymes such as indoleamine 2,3-dioxygenase (IDO) and tryptophan 2,3-dioxygenase (TDO) convert TRP into KYN. The pathway generates metabolites like the kynurenic acid, which has neuroprotective effects, and quinolinic acid, which becomes neurotoxic when accumulated to excess.Serotonin Pathway: TRP is converted into 5-HT, a neurotransmitter involved in regulating mood, sleep, and behavior. 5-HT is further broken down into 5-hydroxyindoleacetic acid (5-HIAA), a marker of serotonin turnover.Indole Pathway: The gut microbiota plays a significant role in converting TRP into indole derivatives such as the indole-3-acetic acid (IAA), indole-3-lactic acid (ILA), and indole-3-propionic acid (IPA). These metabolites are crucial for gut health and regulating the gut–brain axis.

These pathways are central to producing neuroactive metabolites that influence behavior, immune function, and brain development, all of which are relevant to ASD [17,18]. Graphical presentation of the main pathways of tryptophan metabolism, including its conversion to KYN, melatonin, and other metabolites, is presented in Figure 1. This figure combines all three major pathways of TRP metabolism, along with their key metabolites, into a unified overview.

**Table 1 ijms-26-02254-t001:** Key Metabolite Pathways.

Pathway	Key Enzymes	Key Metabolites	Function	Relevance to ASD
**Kynurenine Pathway**	Tryptophan 2,3-Dioxygenase (TDO), Indoleamine 2,3-Dioxygenase (IDO)	Kynurenine (KYN) Kynurenic acid Quinolinic acid	Neuroactive Metabolites immune regulation, neuroinflammation	Altered kynurenine pathway metabolism proposed to modulate neurotransmission and oxidative stress, contributing to ASD symptoms [2].
**Serotonin Pathway**	Tryptophan Hydroxylase	Serotonin (5-HT) 5-Hydroxyindoleacetic acid (5-HIAA)	Mood regulation, behavior, sleep	Early life disruptions of 5-HT synthesis, transport and/or metabolism hypothesized to contribute to mood and behavioral symptoms [19,20,21,22]
**Indole** **Pathway**	Gut Microbiota	Indole-3-acetic acid (IAA) Indole-3-lactic acid (ILA) Indole-3-propionic acid (IPA)	Gut health, antioxidant properties, immune modulation	Altered gut microbiota in ASD affects indole production, impacting gut–brain interactions and neurodevelopment [23].

The urinary levels of TRP metabolites and their ratios have been studied to investigate the possible alterations in tryptophan metabolism and gut microbiota in children with ASD, compared to healthy children [24] (Table 2).

In this study, we aimed to determine whether the urinary concentrations of tryptophan pathway metabolites and their ratios in children with ASD differ from those of the healthy members of the control group. We employed an assay that measures more intermediates than those commonly used [25]. As a result, more ratios can be computed than previously possible, which may provide more insight into the association between TRP metabolism and ASD.

TRP/IAA (tryptophan/indole-3-acetic acid): A higher ratio that would suggest that there is more tryptophan than IAA, which could be a result of the gut microbiota converting tryptophan more slowly. Since IAA is a gut-derived molecule associated with bacterial digestion of tryptophan, it may indicate decreased microbial activity or dysbiosis in ASD. A higher TRP/IAA ratio may indicate compromised gut–brain signaling pathways in ASD, which could affect behavior and neurodevelopment.

TRP/5-HIAA (tryptophan/5-hydroxyindoleacetic acid): A ratio that shows the amount of TRP in relation to the 5-HT turnover rate. Lower 5-HT metabolism may be indicated by a larger TRP/5-HIAA ratio, which could be brought about by either a less enzymatic conversion to 5-HIAA or a lesser 5-HT synthesis. Since 5-HT is essential for mood control and anxiety, this imbalance may be linked to these conditions in ASD.

TRP/NAcTRP (N-acetyltryptophan): TRP may be less efficiently transferred into other metabolic pathways if the TRP/NAcTRP ratio is larger. This could point to a change in tryptophan’s alternate metabolic pathways in ASD, which could have an impact on pathways related to neuroprotection and immunological control.

TRP/IPA (tryptophan/indole-3-propionic acid) and TRP/IAM (tryptophan/indole-3-acetamine): A lower conversion of tryptophan to these neuroactive and antioxidant compounds—which are frequently produced by beneficial gut bacteria—would be indicated by higher TRP/IPA and TRP/IAM ratios. Higher TRP/IPA and TRP/IAM ratios in ASD may indicate decreased antioxidant synthesis from the microbiota, which could exacerbate oxidative stress and neuroinflammation.

TRP/ MIA (tryptophan/methyl indole-3-acetate): A higher ratio of TRP to MIA may indicate a possible decrease in methylation activity, which is frequently engaged in neurotransmitter modulation and detoxification. Increased TRP/MIA ratios in ASD may be a marker of compromised methylation, which could impact gut–brain communication and have an impact on behavioral outcomes.

## 2. Materials and Methods

### 2.1. Subjects

The study population consisted of 88 children, 44 with ASD (36 boys and 8 girls) and 44 healthy children (23 boys and 21 girls), who were members of the same family. In the ASD group, the subjects’ average age was 10.7 years in the range of 4.9–17.0 years. The control group included 44 neurotypical children (siblings), without any acute or chronic illness, who were on average 9.4 years of age, in the range of 0.9–16.7 years. Basic demographic data on patients and control group are shown in Table 3. None of the children received any supplementary vitamin or magnesium intake. Children in the study group were diagnosed with ASD by either an expert pediatrician or a neuropsychiatrist in collaboration with a psychologist (Appendix A). The diagnosis was made using a multidisciplinary approach, which combined a clinical evaluation with a psychological assessment. Children were grouped according to the criteria detailed and summarized by DSM-5 [26]. Additional behavioral ratings were based on a standardized behavior classification for children with ASD, developed by the local educational authority to provide additional school support [27,28,29] and on Childhood Autism Rating Scale (C.A.R.S.) [30]. The CARS questionnaire was filled out by all parents together with a psychologist. The interpretation of CARS scores in our study is as follows:

Score below 28: Typically, a score below 28 indicates that the child is not likely to have ASD. However, it is important to consider other factors and use clinical judgment when making the final diagnosis.

Score between 28 and 36: This range suggests the possibility of mild to moderate autism symptoms. Further evaluation and observation may be needed to determine if the child meets the criteria for ASD.

Score above 36.5: Scores above 36.5 indicate a higher likelihood of significant autism symptoms. It suggests a stronger possibility of the child meeting the diagnostic criteria for ASD.

### 2.2. Collection of Urine Samples

15 mL of morning urine samples were collected into sterile urine containers and stored at −80 °C until analysis.

### 2.3. Metabolite Analysis

#### 2.3.1. Tryptophan Metabolite Analysis

TRP metabolite analysis was performed according to the described protocol [25]. In brief, urine samples were removed from −80 °C and thawed on ice for 90 min. A 250 μL urine aliquot was transferred to a deep well plate and dried via vacuum evaporator (GeneVac at a maximum temperature of 40 °C). A 200 μL 80% isopropanol was added to the dried aliquot, sonicated for 30 s, and vortexed (500 rpm, room temperature, 10 min). The extract was centrifuged at 4000 rpm and 150 μL supernatant transferred to a new deep well plate. The remaining 50 μL sample was re-extracted following the same procedure, and the 200 μL supernatant was removed and combined with the prior 150 μL (i.e., total volume 350 μL). A 40 μL aliquot of the extract was transferred to a new 96-well plate, dried via centrifugal evaporator, and stored at −80 °C. Prior to the mass spectrometry analysis, the urine extract was reconstituted in 20 μL 5% isopropanol containing 4000 nM [^13^C_6_] indole-3-acetatic acid and 400 nM anthranilic acid [^13^C_6_]), vigorously vortexed, and centrifuged for 10 min at 4000 RPM. In addition, for the analysis of tryptophan and kynurenine, urine extracts were diluted 500×, and 40 μL diluted urine transferred to 96-well plate, dried via centrifugal evaporator, and stored at −80 °C. Prior to the mass spectrometry analysis, the diluted urine extract was reconstituted in 20 μL of 5% isopropanol containing 200 nM L-tryptophan [^13^C_11_] [^15^N_2_].

Analysis was performed using a UHPLC system (1260 series Agilent, Santa Clara, CA, USA) coupled with a triple quadrupole mass spectrometer (AJS 6495A, Agilent, Santa Clara, CA, USA). Samples were injected (2 µL) on the analytical column (C18 CSH; 1.7 µm, 2.1 mm i.d. × 100 mm; cat. #186005297; Waters, Milford, MA, USA). The column temperature was held at 40 °C. The mobile phase consisted of solution A (0.1% FA in water) and solution B (0.1% FA in 95% ACN). The flow rate was 300 µL/min. The gradient elution program: 0.0 min 5% B; 5 min 5% B; 10 min 95% B; 11.99 min 95% B; 12 min 5% B; 14 min 5% B. A standard-flow electrospray source operated in positive ion mode (capillary voltage 3.5 kV; gas flow rate 15 L/min at 160 °C; sheath gas flow 12 L/min at 250 °C; nozzle voltage 500 V). A total of 88 transitions were monitored via a dynamic SRM mode analysis, with a 2 min window scheduled around metabolite experimental RT. Additional information about urine analysis is presented in the Appendix A.

#### 2.3.2. Determination of Creatinine in Urine

Urinary creatinine was analyzed on an Advia 2400 analyzer (Siemens Healthcare Diagnostics, Erlangen, Germany) using the Jaffe reaction. Creatinine reacts with picric acid in alkaline medium, the resulting red-colored complex is measured at 505/571 nm.

### 2.4. Statistical Analysis

R version 4.3.1 in conjunction with RStudio version 2023.12.0 was used for the statistical analysis, using the tidyverse suite [31] for visualization and the arsenal package to compare groups. The Shapiro–Wilk tests from the stats package was used to evaluate the distribution of data [32]. Results showed that all datasets had abnormal distributions and, thus, the non-parametric Wilcoxon signed-rank test was employed for comparisons. The Benjamini–Hochberg procedure was used to control false discovery rate, with alpha significance level of 0.05. The differences between the tables generated by the arsenal package and the violin plots are due to missing data, which affects the sample sizes used in each analysis. While the tables provide summary statistics for all available data within each group, the violin plots only display cases with complete data for the specific variable being analyzed. This discrepancy in sample size leads to variations in reported values.

## 3. Results

Our study’s findings indicate that children with ASD have higher tryptophan concentrations (median of 19.94 vs. 16.91; *p* = 0.04) than their siblings. Different patterns of behavior of the median values are noted for the remaining metabolites, but the changes are not statistically significant (Table 4).

KYN, a metabolite of tryptophan degradation pathway, was found at higher levels in children with ASD compared to the healthy children (82.34 vs. 71.20; *p* = 0.86). Although not statistically significant, this pattern is consistent with previous studies showing elevated KYN levels in ASD, which is associated with increased neuroinflammation and immune dysregulation [33,34]. The insignificant increase in the moderately affected children (77.23 vs. 71.20; *p* = 1.00) and a more pronounced rise in the severely affected children (97.70 vs. 71.21; *p* = 0.28) could reflect a greater activation of the kynurenine pathway in more severe cases, further supporting the link between severity and inflammatory responses.

In our study, children with ASD had higher urinary TrpN concentrations compared to the children in the control group (441.1 vs. 389.1; *p* = 0.10), although this difference was not statistically significant. Notably, in the control group, the TrpN levels were insignificantly lower than in children with severe illness (389.1 vs. 391.7; *p* = 0.72), yet, the difference was more pronounced in children with moderate illness (528.1 vs. 389.1; *p* = 0.13), approaching statistical significance. Children with ASD had insignificantly lower 5-HTP concentrations compared to the children in the control group (18.39 vs. 22.21; *p =* 0.15), yet the lower level was statistically significant when only considering children with severe illness (12.46 vs. 22.21; *p =* 0.01). 

Figure 2 displays the findings of the CARS evaluations completed by the children’s parents/guardians, in cooperation with a psychologist, for the group of children with ASD, in comparison to siblings as a control group. The healthy group’s median score was 15.25, whereas in the ASD group it was 36.00.

For each metabolite, correlation with CARS score was tested. Graphical correlations with TRP and KYN are displayed. While the association for TRP is statistically significant, the correlation for KYN is not (Figure 3 and Figure 4).

The data for the remaining parameters can be found in the Appendix A.

Other relationships mainly did not achieve statistical significance despite the trends that were seen; this could be because of sample size restrictions, data variability, or possible confounding variables.

Metabolite concentration ratios were calculated, comparing the children in the control group and the ASD group.

The ratios of KYN/TRP and TRP/5-HTP are displayed in the text (Figure 5 and Figure 6). The Appendix A contains additional graphical presentations.

The ratio TRP/5-HTP is significantly higher in our ASD group (0.62 vs. 1.06; *p* = 0.01) although it is not significant in the moderate (0.69 vs. 0.62; *p* = 0.68) or severe (1.83 vs. 0.62; *p* = 0.09) subgroups. Also, we found statistically significant difference in the ratio TRP/NAcTRP between control group and severe subgroup (0.09 vs. 0.11; *p* = 0.04), and in the ratio TRP/TrpN between the ASD subgroups (0.04 vs. 0.06; *p* = 0.03).

## 4. Discussion

The metabolic pathways of tryptophan are complex and involve both endogenous (host-microbial) and exogenous (dietary) components. Potential disturbances in these pathways are indicated by variations in the metabolite levels and ratios.

Studies suggest that less severe cases of ASD may differ from more severe cases due to different compensatory mechanisms [11,33]. Comparing the entire group yields slightly different findings than dividing our group of children into those with moderate disorder and those with severe disorder and then adding a sibling to each group. We can still make some inferences even though there are only 16 children with severe ASD. The Appendix A contains the comparison results for tryptophan and the various metabolites. In the group with ASD, TRP differs from that of healthy siblings in a statistically significant amount (19.94 vs. 16.91; *p* = 0.04), whereas in the groups with severe (21.37 vs. 16.91; 0.09) and moderate (19.81 vs. 16.91; 0.06) impairment, the difference is statistically negligible. Also, a statistically significant different amount of 5-HTP was observed in children with severe impairment compared to that of healthy children (12.46 vs 22.21; *p* = 0.01). We did not find a statistically significant difference for any of the other metabolites. However, we did find an almost significant difference for TrpN, with an increase in children with moderate impairment (389.13 vs. 528.14; *p* = 0.13). The changes in the median values based on the severity of disease are displayed in Table 4.

Elevated KYN levels in the severe ASD group may indicate enhanced indoleamine 2,3-dioxygenase (IDO) activity, triggered by pro-inflammatory cytokines, a mechanism frequently seen in ASD [11]. These trends highlight the importance of kynurenine pathway dysregulation in ASD pathology, despite the lack of statistical significance in these particular comparisons. Further studies with larger cohorts are needed to help clarify the association of kynurenine pathway dysregulation with ASD pathology and direct measurement of enzymatic activities may be valuable.

For example, assays of indoleamine 2,3-dioxygenase (IDO), which routes tryptophan metabolism into the kynurenine pathway, and tryptophan decarboxylase, which changes TRP into TrpN, are recommended. Furthermore inflammation and neurological abnormalities in ASD have been associated with kynurenine pathway activity [15,33]. The adaptive responses in tryptophan metabolism may depend on the severity of the disorder. For example, tryptophan decarboxylase activity may vary depending on ASD severity, and a and a higher activity could exacerbate neurotoxic effects. Further studies are needed to clarify these enzyme dynamics and their implications for ASD pathophysiology.

The ratios calculated in our study are presented in Table 5. The KYN/TRP (kynurenine/tryptophan) ratio is a widely recognized marker of tryptophan metabolism and immune activation, as elevated kynurenine levels are often triggered by inflammation-induced indoleamine 2,3-dioxygenase (IDO) activity [11]. In our study, the KYN/TRP ratio was insignificantly higher in the ASD group (4.21 vs. 4.67; *p* = 0.96). Although the difference was not statistically significant, this trend aligns with prior studies reporting an increased kynurenine pathway activation in ASD [34]. Therefore, our findings are consistent with the general pattern observed in ASD research, even though the lack of statistical significance limits the strength of this conclusion. Similarly, the ratio remained stable in the moderate group (4.21 vs. 4.60; *p* = 0.47) but increased slightly in the severe group (4.21 vs. 5.53; *p* = 0.27). Although these changes are not statistically significant, the higher ratio in the severe group could indicate heightened immune activation and neuroinflammation, which has been linked to a greater severity of ASD symptoms [33]. The KYN/TRP ratio’s rise with the increasing severity may point to an inflammation-driven shift in tryptophan metabolism toward the kynurenine pathway, reinforcing the role of immune dysregulation in ASD pathology.

The significantly higher TRP/5-HTP ratio in our ASD group (0.62 vs. 1.06; *p* = 0.01) and reduced 5-HTP concentrations observed (18.39 vs. 22.21; *p* = 0.15) is consistent with prior literature that indicates serotonin dysregulation in ASD [11,35]. Although not statistically significant, the tryptophan to 5-HTP ratio is slightly higher in the moderate subgroup (0.69 vs. 0.62; *p* = 0.68) and shows a statistically almost significant increase in the severe group (1.83 vs. 0.62; *p* = 0.09). Moreover, an insignificant lower 5-HTP concentrations were observed in moderate ASD group compared to control group (21.52 vs 22.21; *p* = 0.40), and a significant difference in severe ASD group (12.46 vs 22.21 vs.; *p* = 0.01). This trend aligns with the broader hypothesis that serotonin synthesis deficits contribute to ASD-related behaviors, particularly in individuals with more severe forms of the disorder [33], with more pronounced disruption in the serotonin synthesis pathway as symptom severity worsens. The marked decrease in 5-HTP concentrations in the severe group warrants further investigation to clarify the role of serotonin dysregulation in more severe forms of ASD.

Table 6 and Table 7 illustrate the differences in metabolite and dimensional change patterns between mild and severe disorders. This indicates that moderate and severe diseases entail distinct pathways. 

A less varied pattern of these ratios, which are shown in Table 7, is obtained when we examine the computed ratios between particular metabolites.

In some cases, we found significant variations between the two (↑↓,↓↑), and those statuses are displayed in column (3–4) of Appendix A.

## 5. Conclusions

The metabolism of tryptophan is a complex process that involves multiple pathways and enzymes, and the balance between these pathways may play a role in various physiological and pathological conditions, including in the autism spectrum disorder (ASD). However, the proportion of each pathway in healthy individuals and individuals with ASD may vary, depending on various, factors such as age, sex, diet, and genetic background.

Our findings suggest that the dysregulation of the kynurenine pathway and the resulting increase in kynurenine levels may be a common feature of ASD. However, our results also show that the dysregulation of other tryptophan metabolites may also contribute to the pathophysiology of ASD and that further research is needed to fully understand the role of tryptophan metabolism in this disorder.

The most common way to express the results of studies investigating tryptophan metabolism in healthy individuals versus those with ASD is through the use of ratios. Ratios are a useful way to compare the levels of different metabolites because they consider variations in the total amount of tryptophan available for metabolism.

The most commonly used ratio in these studies is the KYN/TRP ratio, which is the ratio of kynurenine to tryptophan levels in blood, plasma, or urine. This ratio is thought to reflect the activity of the kynurenine pathway, which is one of the major pathways of tryptophan metabolism. In our study, in addition to the KYN/TRP ratio, other ratios involving tryptophan metabolites were present in urine samples from individuals with ASD.

We have to be aware that while ratios are a useful way to compare the levels of different metabolites, they do not provide information about their absolute levels. Therefore, it is important to interpret the ratios in conjunction with absolute metabolite levels to gain a more complete understanding of the metabolic changes associated with ASD.

For the control group, we employed the siblings of a group of children with ASD, which was crucial for this study. Nevertheless, the number of cases and the ability to interpret power decline when we split the group into individuals with mild and severe ASD.

Although the calculated ratios, like KYN/TRP and TRP/5-HTP are helpful markers of metabolic changes in tryptophan pathways, there is still a lack of research on how they might be directly applied to ASD. Because there are no defined baseline values for these ratios in children with ASD compared to healthy children, interpretation becomes more difficult. To put these findings in context, assess their clinical relevance, and create normative data more research is needed.

Future studies should aim to systematically measure a broader spectrum of tryptophan metabolites across larger cohorts, both in ASD and control groups, to better understand the role of these metabolic pathways and to validate their clinical relevance.

To ensure a comprehensive analysis, we documented recent antibiotic use and significant comorbidities within our study sample. However, these factors were excluded from the current analysis to maintain the focus on the broader metabolic trends. Comorbid conditions, such as gastrointestinal disorders prevalent among children with ASD, and the potential impact of antibiotics on gut microbiota and subsequent metabolic profiles are recognized as critical variables in biomarker research.

Future investigations will aim to integrate these factors by gathering detailed data on antibiotic treatments and comorbidities. This approach will enable a more nuanced understanding of how such variables interact with tryptophan metabolism and influence urinary metabolite profiles. By collaborating with multidisciplinary teams and expanding the sample size, we aim to strengthen the robustness and clinical relevance of our findings.

## Figures and Tables

**Figure 1 ijms-26-02254-f001:**
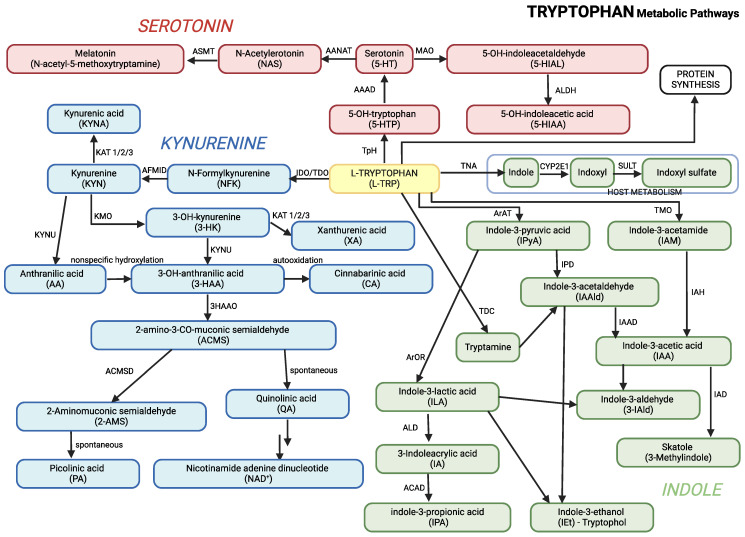
Main pathways of TRP metabolism. 3-HAAO: 3-Hydroxyanthranilic acid 3,4-dioxygenase, AAAD: Aromatic Amino Acid Decarboxylase, AANAT: Aralkylamine N-Acetyltransferase, ACD: Acyl-CoA dehydrogenase, ACMSD: 2-amino-3-carboxymuconate-6-semialdehyde decarboxylase, AFMID—Kynurenine formamidase, ALD: (R)-3-(aryl)lactoyl-CoA dehydratase, ALDH: Aldehyde dehydrogenase, ArAT: Aromatic amino acid aminotransferase, ArOR: Aromatic 2-oxoacid reductase, ASMT: Acetylserotonin O-Methyltransferase, CYP2E1: Cytochrome P450 2E1, IAH: Indolacetamide hydrolase, IAAD: Indoleacetaldehyde oxidase, IAD: Indoleacetate decarboxylase, IDO: Indoleamine 2,3-Dioxygenase, IPD: Indolepyruvic decarboxylase, KAT: Kynurenine aminotransferases 1/2/3, KMO: Kynurenine 3-Monooxygenase, KYNU: Kynureninase, MAO: Monoamine Oxydase, SULT: Sulfotransferase, TDO: Tryptophan 2,3-Dioxygenase, TMO: Tryptophan 2-Monooxygenase, TNA: Tryptophanase, TpH: Tryptophan Hydroxylase, TDC: Tryptophan Decarboxylase.

**Figure 2 ijms-26-02254-f002:**
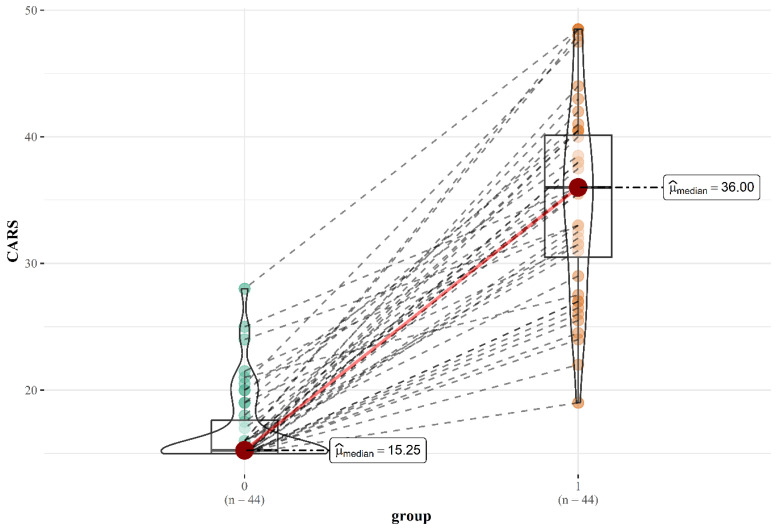
CARS scores for the children with ASD and the control group.

**Figure 3 ijms-26-02254-f003:**
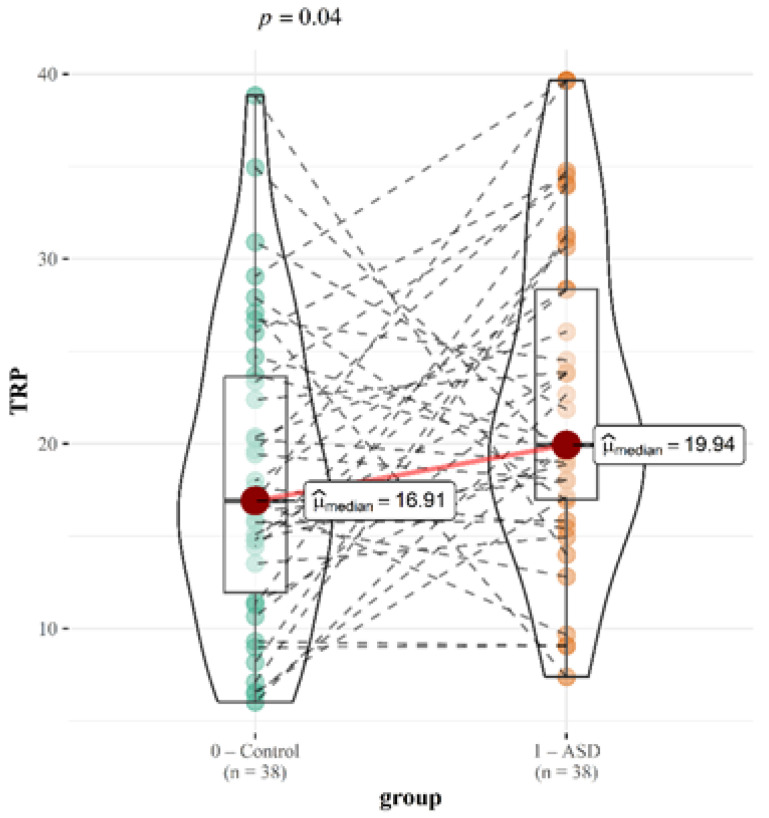
Graphical presentation of the urinary TRP results. Median values and calculated *p* values are presented for both groups.

**Figure 4 ijms-26-02254-f004:**
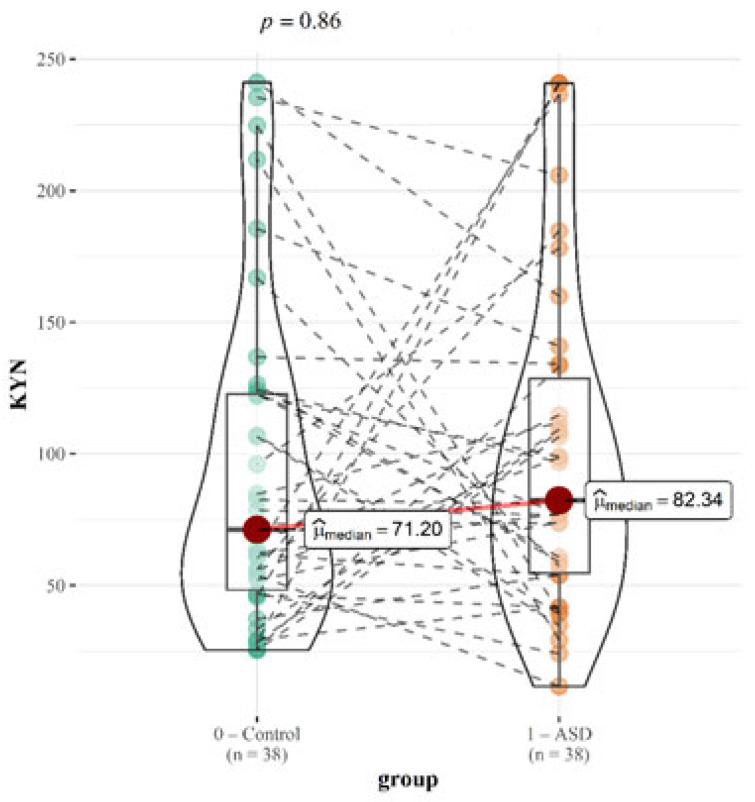
Graphical presentation of the urinary KYN results. Median values and calculated *p* values are presented for both groups.

**Figure 5 ijms-26-02254-f005:**
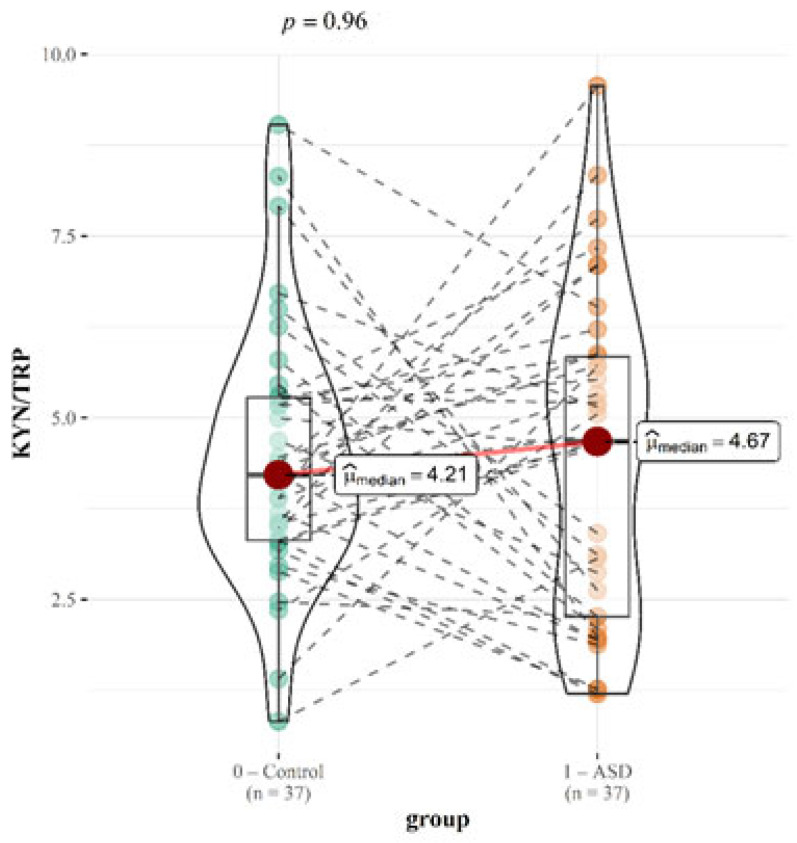
Graphical presentation of the results of the urinary KYN/TRP ratio. Median values and calculated *p* values are presented for both groups.

**Figure 6 ijms-26-02254-f006:**
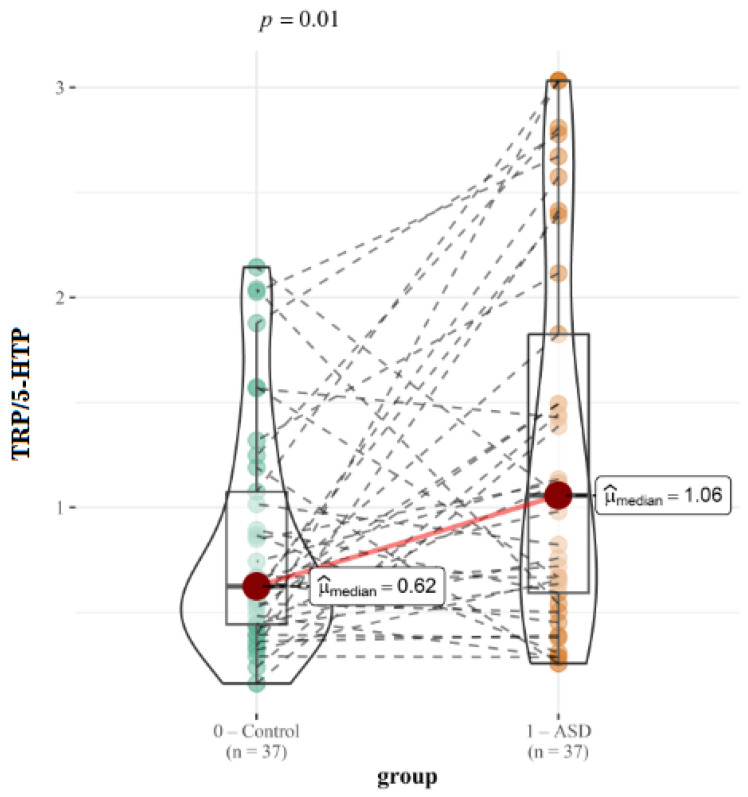
Graphical presentation of the results for urinary TRP/5-HTP ratio. Median values and calculated *p* values are presented for both groups.

**Table 2 ijms-26-02254-t002:** Commonly calculated ratios and their putative relevance to ASD.

Ratio	Description	Hypothesized relevance to ASD
**KYN/TRP (kynurenine/tryptophan)**	Reflects the activity of the kynurenine pathway.	Elevated KYN/TRP ratios in ASD suggest increased kynurenine pathway activity, linked to neuroinflammation.
**TRP/IAA (tryptophan/** **indole-3-acetic acid)**	Indicates the activity of the indole pathway.	Reduced indole derivatives in ASD reflect gut microbiota alterations, affecting gut–brain axis signaling.
**5-HIAA/5-HTP (5-hydroxyindoleacetic acid/5-hydroxytryptophan)**	Reflects serotonin turnover and synthesis efficiency.	Decreased serotonin turnover in ASD is linked to behavioral and mood disturbances.
**TRP/IPA (tryptophan/** **indole-3-propionic acid)**	Reflects antioxidant production via the gut microbiota.	Decreased IPA in ASD indicates reduced antioxidant defense and gut dysbiosis.
**TRP/TrpN (tryptophan/tryptamine)**	Reflects serotonin and melatonin precursor production.	Lower TrpN levels in ASD suggest impaired serotonin and melatonin synthesis.

**Table 3 ijms-26-02254-t003:** Basic demographic data on patients and control group.

	0 (N = 44)	1 (N = 44)
**Control (0), Patients (1)**		
Male (M)	23 (52.3%)	36 (81.8%)
Female (F)	21 (47.7%)	8 (18.2%)
**Age**		
Mean (SD)	9.36 (3.51)	10.68 (3.19)
Median (Q1, Q3)	9.35 (6.80, 11.38)	10.70 (8.38, 13.00)
Min—Max	0.90–16.70	4.90–17.00

**Table 4 ijms-26-02254-t004:** Tryptophan metabolites in the urine of the control group (1) and the ASD group (2), a group with CARS score up to 36 (3) and with CARS > 36.5 (4), along with *p* values; all values are in nM/mmol of creatinine.

Metabolite	Control (N = 44)	ASD (N = 44)	CARS < 36 (N = 28)	CARS > 36.5 (N = 16)	*p* Value			
	**1**	**2**	**3**	**4**	**1–2**	**1–3**	**1–4**	**3–4**
Tryptophan–TRP					0.04	0.06	0.09	0.74
Mean (SD)	18.20 (8.17)	22.30 (8.53)	22.96 (9.51)	21.74 (6.52)				
Median (Q1, Q3)	16.91 (11.97, 23.63)	19.94 (16.99, 28.36)	19.81 (16.16, 33.22)	21.37 (19.29, 25.67)				
Min–Max	6.03–38.83	7.38–39.65	9.07–39.65	7.38–31.30				
Anthranilate–ATA					0.99	0.50	0.22	0.40
Mean (SD)	21.21 (9.43)	21.09 (10.78)	23.78 (13.10)	17.34 (7.61)				
Median (Q1, Q3)	21.44 (14.95, 25.43)	19.86 (13.90, 28.61)	20.55 (13.97, 32.92)	19.29 (10.24, 21.16)				
Min–Max	5.52–50.67	4.52–50.73	5.04–53.00	4.52–29.51				
Indole-3-aceticacid–IAA					0.78	0.75	0.74	0.83
Mean (SD)	1.79 (1.48)	1.86 (1.37)	1.97 (1.49)	1.57 (1.07)				
Median (Q1, Q3)	1.32 (0.70, 2.28)	1.67 (0.88, 2.61)	1.68 (0.83, 2.76)	1.18 (0.88, 2.37)				
Min–Max	0.11–5.84	0.20–5.80	0.20–5.80	0.22–3.53				
Indole-3-aldehyde–IALD					0.10	0.45	0.19	0.50
Mean (SD)	200.29 (119.30)	165.20 (83.91)	172.01 (88.14)	152.50 (76.64)				
Median (Q1, Q3)	174.38 (102.35, 287.53)	147.13 (104.24, 223.62)	147.13 (111.80, 226.26)	134.39 (91.37, 188.69)				
Min–Max	17.43–455.15	30.44–358.36	30.44–358.36	62.03–313.85				
Indole-3-acetamide–IAM					1.00	0.91	0.96	0.78
Mean (SD)	61.77 (37.16)	61.78 (36.77)	65.64 (41.31)	57.77 (28.18)				
Median (Q1, Q3)	55.71 (31.43, 79.18)	57.14 (30.33, 79.70)	57.14 (32.71, 85.72)	58.03 (32.09, 79.70)				
Min–Max	10.43–160.04	12.85–160.99	18.11–160.99	12.85–100.63				
Indole-3-butyricacid–IBA*					0.98	0.89	0.57	0.95
Mean (SD)	20.13 (17.40)	20.53 (18.82)	21.76 (20.33)	15.77 (13.97)				
Median (Q1, Q3)	17.20 (8.71, 23.79)	12.97 (8.10, 33.69)	12.51 (7.00, 37.91)	12.21 (8.17, 18.71)				
Min–Max	0.81–75.60	0.03–68.17	0.03–68.17	0.40–44.33				
Indole-3-lacticacid–ILA					0.53	0.19	0.47	0.96
Mean (SD)	1.08 (0.58)	0.99 (0.58)	0.94 (0.52)	0.94 (0.51)				
Median (Q1, Q3)	1.01 (0.61, 1.40)	0.75 (0.59, 1.13)	0.74 (0.67, 1.02)	0.83 (0.52, 1.40)				
Min–Max	0.18–2.41	0.27–2.67	0.28–2.39	0.27–1.80				
Indole-3-propionicacid–IPA					0.27	0.35	0.16	0.94
Mean (SD)	5.13 (3.04)	4.16 (3.11)	4.40 (3.60)	3.57 (1.28)				
Median (Q1, Q3)	4.53 (3.42, 6.86)	3.16 (2.08, 5.15)	3.64 (1.91, 5.54)	3.16 (2.75, 4.50)				
Min–Max	0.20–13.98	0.00–13.11	0.00–13.11	1.27–5.71				
Kynurenine–KYN					0.86	1.00	0.28	0.62
Mean (SD)	92.94 (61.63)	99.77 (61.68)	80.67 (45.97)	99.77 (46.39)				
Median (Q1, Q3)	71.20 (48.30, 122.81)	82.34 (54.76, 128.66)	77.23 (47.82, 105.30)	97.70 (64.36, 134.02)				
Min–Max	25.41–241.15	11.62–240.85	11.62–205.89	29.30–184.66				
Methylindole-3-acetate–MIA*					0.80	0.64	0.86	0.13
Mean (SD)	0.07 (0.04)	0.08 (0.06)	0.08 (0.05)	0.06 (0.04)				
Median (Q1, Q3)	0.06 (0.04, 0.10)	0.07 (0.04, 0.11)	0.07 (0.03, 0.11)	0.06 (0.04, 0.07)				
Min–Max	0.01–0.21	0.00–0.23	0.00–0.20	0.00–0.14				
N-acetyl-tryptophan–NAcTRP					0.92	0.54	0.75	0.26
Mean (SD)	206.05 (118.34)	200.64 (83.98)	212.55 (87.76)	178.41 (74.11)				
Median (Q1, Q3)	185.08 (95.68, 279.38)	185.54 (149.26, 252.54)	225.92 (159.52, 265.84)	184.49 (112.53, 217.93)				
Min–Max	35.88–488.19	56.47–363.64	56.47–344.29	80.06–363.64				
Tryptamine–TrpN					0.10	0.13	0.72	0.11
Mean (SD)	389.89 (209.89)	456.33 (257.72)	518.74 (278.95)	365.38 (173.84)				
Median (Q1, Q3)	389.13 (246.22, 483.64)	441.10 (230.08, 649.44)	528.14 (281.69, 763.43)	391.67 (238.62, 451.97)				
Min–Max	56.29–1041.14	63.19–1075.56	63.19–1075.56	125.02–653.10				
5-hydroxyindoleaceticacid–5HIAA					0.61	0.28	0.63	0.17
Mean (SD)	48.77 (27.63)	50.76 (25.04)	53.62 (23.03)	44.82 (28.83)				
Median (Q1, Q3)	39.12 (31.35, 61.84)	48.64 (31.03, 61.78)	48.64 (33.60, 63.86)	42.98 (23.75, 49.14)				
Min–Max	17.96–116.40	9.74–116.26	24.96–101.83	9.74–116.26				
5-hydroxy-L-tryptophan–5-HTP					0.15	0.40	0.01	0.14
Mean (SD)	22.79 (12.56)	19.13 (12.76)	21.98 (14.30)	14.24 (7.92)				
Median (Q1, Q3)	22.21 (11.71, 30.28)	18.39 (10.09, 24.31)	21.52 (12.10, 26.27)	12.46 (9.79, 19.67)				
Min–Max	0.71–49.81	0.79–54.70	3.48–54.70	0.79–28.73				
5-Methoxyindoleacetate–5MIAA *					0.89	0.13	0.37	0.19
Mean (SD)	15.12 (9.89)	16.40 (13.23)	11.43 (9.37)	19.68 (13.52)				
Median (Q1, Q3)	15.11 (6.53, 22.07)	10.93 (7.69, 22.06)	9.05 (6.86, 11.35)	16.73 (7.84, 31.74)				
Min–Max	0.79–45.27	1.12–52.02	0.90–38.93	4.06–44.47				
U-Creatinine					0.43	0.68	0.50	0.99
Mean (SD)	7.58 (4.27)	8.36 (4.76)	8.40 (5.06)	8.29 (4.33)				
Median (Q1, Q3)	6.55 (4.62, 9.93)	7.70 (4.55, 10.88)	7.10 (4.40, 10.65)	8.40 (5.25, 11.00)				
Min–Max	0.10–18.30	0.70–18.20	2.30–18.20	0.70–15.90				

* We have performed the calculation, but due to low detection frequency we did not interpret it.

**Table 5 ijms-26-02254-t005:** Variations of the tryptophan metabolite ratios in the urine of the control group (1) and the ASD group (2), a group with CARS scores up to 36 (3) and with CARS > 36.5 (4), along with *p* values.

Metabolite Ratios	Control (N = 44)	ASD (N = 44)	CARS < 36 (N = 28)	CARS > 36.5 (N = 16)	*p* Value			
	**1**	**2**	**3**	**4**	**1–2**	**1–3**	**1–4**	**3–4**
KYN/TRP					0.96	0.47	0.27	0.17
Mean (SD)	4.47 (1.76)	4.48 (2.20)	4.20 (1.93)	5.39 (2.81)				
Median (Q1, Q3)	4.21 (3.32, 5.28)	4.67 (2.27, 5.84)	4.60 (2.53, 5.42)	5.53 (3.57, 6.93)				
Min–Max	0.82–9.03	1.20–9.56	1.20–8.34	1.27–10.41				
KYN/ATA					0.22	0.54	0.14	0.82
Mean (SD)	4.07 (3.14)	4.72 (2.69)	4.34 (2.46)	6.93 (4.63)				
Median (Q1, Q3)	2.71 (1.98, 5.61)	4.53 (2.83, 5.59)	4.46 (2.83, 5.41)	5.30 (3.41, 8.66)				
Min–Max	0.30–12.54	0.57–12.81	0.57–9.85	2.34–17.01				
TRP/IAA					0.34	0.86	0.50	0.29
Mean (SD)	15.32 (10.77)	13.71 (8.80)	13.14 (5.82)	20.60 (19.27)				
Median (Q1, Q3)	11.63 (6.73, 22.28)	11.85 (7.14, 17.96)	11.85 (9.19, 18.12)	13.19 (6.16, 32.42)				
Min–Max	2.12–40.36	3.09–44.43	3.09–22.02	3.44–63.91				
IAA/IALD					0.36	0.60	0.70	1.00
Mean (SD)	0.01 (0.01)	0.01 (0.01)	0.01 (0.01)	0.01 (0.00)				
Median (Q1, Q3)	0.01 (0.00, 0.01)	0.01 (0.01, 0.01)	0.01 (0.01, 0.01)	0.01 (0.01, 0.01)				
Min–Max	0.00–0.03	0.00–0.02	0.00–0.03	0.00–0.02				
TRP/IAM					0.42	0.86	0.28	0.27
Mean (SD)	0.39 (0.22)	0.45 (0.32)	0.42 (0.33)	0.53 (0.30)				
Median (Q1, Q3)	0.34 (0.23, 0.50)	0.31 (0.20, 0.76)	0.27 (0.17, 0.73)	0.44 (0.29, 0.79)				
Min–Max	0.10–1.03	0.02–1.08	0.02–1.05	0.19–1.08				
TRP/ILA					0.24	0.06	0.56	0.81
Mean (SD)	21.16 (13.25)	25.38 (14.16)	26.08 (14.03)	23.81 (14.36)				
Median (Q1, Q3)	17.41 (13.06, 27.07)	22.58 (14.49, 32.35)	25.71 (14.21, 32.35)	19.01 (15.41, 29.69)				
Min–Max	4.32–57.27	0.79–56.88	0.79–56.88	5.88–54.20				
TRP/NAcTRP					0.31	0.07	0.04	0.43
Mean (SD)	0.11 (0.07)	0.12 (0.04)	0.12 (0.04)	0.13 (0.07)				
Median (Q1, Q3)	0.09 (0.07, 0.15)	0.12 (0.10, 0.15)	0.13 (0.10, 0.15)	0.11 (0.09, 0.16)				
Min–Max	0.02–0.26	0.05–0.25	0.05–0.22	0.06–0.31				
TRP_TRPN					0.46	0.39	0.37	0.03
Mean (SD)	0.05 (0.03)	0.05 (0.02)	0.04 (0.02)	0.05 (0.02)				
Median (Q1, Q3)	0.04 (0.03, 0.07)	0.05 (0.03, 0.06)	0.04 (0.03, 0.05)	0.06 (0.05, 0.06)				
Min–Max	0.01–0.13	0.01–0.09	0.02–0.09	0.01–0.09				
TRP/5-HIAA					0.98	0.44	0.92	0.82
Mean (SD)	0.46 (0.27)	0.48 (0.33)	0.39 (0.21)	0.62 (0.48)				
Median (Q1, Q3)	0.37 (0.27, 0.63)	0.33 (0.22, 0.57)	0.33 (0.23, 0.54)	0.41 (0.21, 1.01)				
Min–Max	0.06–1.17	0.13–1.32	0.13–0.98	0.17–1.72				
TRP/5-HTP					0.01	0.68	0.001	0.08
Mean (SD)	0.83 (0.55)	1.28 (0.89)	0.84 (0.54)	1.69 (0.83)				
Median (Q1, Q3)	0.62 (0.44, 1.07)	1.06 (0.59, 1.83)	0.69 (0.40, 1.12)	1.83 (1.06, 2.39)				
Min–Max	0.16–2.14	0.26–3.03	0.26–2.41	0.54–2.81				
TRP/IPA					0.41	0.74	0.43	0.67
Mean (SD)	4.20 (2.86)	4.86 (3.38)	4.81 (3.50)	1.41 (1.09)				
Median (Q1, Q3)	3.83 (2.06, 5.91)	3.81 (1.99, 6.89)	3.73 (1.89, 6.86)	1.15 (0.64, 1.82)				
Min–Max	0.09–10.29	0.64–13.18	0.72–13.18	0.27–4.05				
IAA/KYN					0.81	0.61	0.67	0.67
Mean (SD)	0.02 (0.01)	0.02 (0.02)	0.02 (0.02)	0.02 (0.01)				
Median (Q1, Q3)	0.02 (0.01, 0.03)	0.02 (0.01, 0.03)	0.02 (0.01, 0.03)	0.02 (0.00, 0.02)				
Min–Max	0.00–0.06	0.00–0.06	0.00–0.06	0.00–0.05				
KYN/IALD					0.19	0.12	0.44	0.93
Mean (SD)	0.55 (0.44)	0.68 (0.45)	0.74 (0.49)	0.68 (0.51)				
Median (Q1, Q3)	0.38 (0.28, 0.72)	0.60 (0.40, 0.77)	0.61 (0.40, 0.93)	0.60 (0.45, 0.65)				
Min–Max	0.06–1.73	0.05–1.66	0.05–1.66	0.13–1.93				

**Table 6 ijms-26-02254-t006:** Metabolite alterations by degree of impairment in comparison to siblings.

Moderate	Severe	Metabolite
↑	↑	TRP, 5-HIAA, Creatinine
↑	↓	IAA, IAM, NAcTRP, TrpN
↓	↑	ATA, ILA, IPA, KYN,
↓	↓	IALD, 5-HTP

**Table 7 ijms-26-02254-t007:** Metabolite ratios alterations by degree of impairment in comparison to siblings.

Moderate	Severe	Metabolite
↑	↑	KYN/TRP, KYN/ATA, TRP/IAA, TRP/NAcTRP, TRP/5-HTP, TRP/IPA, KYN/IALD
↑	↓	TRP/ILA
↓	↑	TRP/IAM
↓	↓	TRP/5-HIAA

## Data Availability

The data of LC MRM analysis are deposited at Panorama Public (https://panoramaweb.org/trh0XH.url, accessed on 12 October 2024).

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
