# Peer review of "Urinary Metabolomic Profile in Children with Autism Spectrum Disorder"

_ijms, 2025, doi:10.3390/ijms26052254_

Round 1
Reviewer 1 Report
Comments and Suggestions for Authors
The authors presented a nice and original manuscript entitled “Urinary metabolomic profile in children with Autism Spectrum Disorder”, sent for publication in the International Journal of Molecular Sciences. The authors evaluated the potential involvement of tryptophan metabolism and its metabolites as biomarkers related to Autism Spectrum Disorder (ASD) clinical progression and severity. The Introduction of the manuscript has high quality of content, and Tables 1 and 2 and Figure 1 are of great interest to the reader as they summarize a complex theme in a practical way. Some points should be evaluated by the authors:
1. I agree with the authors that there is probably a different activation of metabolic pathways involved in tryptophan metabolism. Previous studies had conflicting findings and there are several points of concerns, including the heterogeneous genetic and metabolic basis involved with ASD. Several Inborn Errors of Metabolism have been associated with both intellectual disability and ASD (Senarathne et al., 2023; Zigman et al., 2021; Inci et al., 2021; Kiykim et al., 2016; Ghaziuddin et al., 2013), even in cases with nonsyndromic ASD (Campistol et al., 2016). A key question is if the authors have any information about the genetic basis involved with ASD cases studied in their sample.
2. The use of cofactors and vitamin supplements is quite common in patients with ASD during their treatment. It would be interesting if the authors describe if they have evaluated previously the recent use of such therapies in the studied group. The same aspect is valid for antimicrobial therapy, as the use of antibiotics, for example, may lead to change of microbiota, which may be involved with the increased biosynthesis of tryptophan and related metabolites.
3. There is a minor typo in line 256 which needs to be corrected: “depostied”.
4. Why do the authors think the urinary tryptamine concentration and kynurenine levels in controls and ASD children were not markedly and significantly different? Could this be due to the relatively small size of the studied sample?
Author Response
1.I agree with the authors that there is probably a different activation of metabolic pathways involved in tryptophan metabolism. Previous studies had conflicting findings and there are several points of concerns, including the heterogeneous genetic and metabolic basis involved with ASD. Several Inborn Errors of Metabolism have been associated with both intellectual disability and ASD (Senarathne et al., 2023; Zigman et al., 2021; Inci et al., 2021; Kiykim et al., 2016; Ghaziuddin et al., 2013), even in cases with nonsyndromic ASD (Campistol et al., 2016). A key question is if the authors have any information about the genetic basis involved with ASD cases studied in their sample.
Because ASD is a complex disorder influenced by both genetic and environmental risk factors, we specifically selected families for our study in which a genetic background of the disorder had been ruled out. This approach was intended to minimize genetic variability and allow us to focus on other potential contributors, such as metabolic pathways. By excluding cases with a known genetic basis, we aimed to reduce confounding factors and enhance the specificity of our findings related to tryptophan metabolism in ASD.
- The use of cofactors and vitamin supplements is quite common in patients with ASD during their treatment. It would be interesting if the authors describe if they have evaluated previously the recent use of such therapies in the studied group. The same aspect is valid for antimicrobial therapy, as the use of antibiotics, for example, may lead to change of microbiota, which may be involved with the increased biosynthesis of tryptophan and related metabolites.
We appreciate the reviewer’s observation regarding the potential impact of comorbidities and antibiotic therapies on metabolic profiles in children with ASD. These aspects are crucial considerations in biomarker research. We documented recent antibiotic use and significant comorbidities in our sample. However, these factors were excluded from the current analysis. We acknowledge that this topic needs more research, and we plan to include these factors in subsequent investigations.
Data on comorbid illnesses, like gastrointestinal disorders, which are prevalent in children with ASD, could help elucidate how these factors interact with tryptophan metabolism. Similarly, the use of antibiotics, which affect gut flora, may significantly influence urinary metabolite profiles and warrants further exploration..
The goal of future research will be to gather and examine comprehensive data on antibiotic treatments and comorbidities. By collaborating with multidisciplinary teams and conducting studies on larger sample sizes, we aim to deepen our understandingof how these factors affect metabolic profiles in ASD. To improve the robustness and therapeutic relevance of our findings, we are dedicated to resolving these constraints in subsequent research. Research on metabolic biomarkers in ASD could be greatly advanced by a deeper comprehension of the function of comorbidities and antibiotic use.
The following paragraph has been added to the text:
To ensure a comprehensive analysis, we documented recent antibiotic use and significant comorbidities within our study sample. However, these factors were excluded from the current analysis to maintain the focus on broader metabolic trends. Comorbid conditions, such as gastrointestinal disorders prevalent among children with ASD, and the potential impact of antibiotics on gut microbiota and subsequent metabolic profiles are recognized as critical variables in biomarker research.
Future investigations will aim to integrate these factors by gathering detailed data on antibiotic treatments and comorbidities. This approach will enable a more nuanced understanding of how such variables interact with tryptophan metabolism and influence urinary metabolite profiles. By collaborating with multidisciplinary teams and expanding the sample size, we aim to strengthen the robustness and clinical relevance of our findings.
3.There is a minor typo in line 256 which needs to be corrected: “depostied”.
Thank you for reading the text very carefully and finding the error that we missed. We have made a correction in the text.
- Why do the authors think the urinary tryptamine concentration and kynurenine levels in controls and ASD children were not markedly and significantly different? Could this be due to the relatively small size of the studied sample?
We appreciate the reviewer's observation regarding the absence of significant differences in urinary tryptamine and kynurenine levels between the ASD and control groups. This is an important consideration in interpreting our results. The relatively small sample size of our study, although consistent with other exploratory metabolomic research, may have limited our ability to detect small but potentially meaningful differences in tryptamine and kynurenine levels.
Furthermore, the absence of noticeable variations may be due to interindividual heterogeneity in tryptophan metabolism. Factors such as dietary habits, gut microbiota composition, and genetic background significantly influence these pathways, potentially obscuring differences in metabolic profiles. Furthermore, the inherent variability of ASD itself may contribute to the masking of distinct metabolic patterns.
Larger, multicentric cohorts in future research may assist validate these results and offer the statistical strength required to identify small variations. Additionally, combining information on comorbidities, gut microbiota composition, and diet may help clarify the underlying mechanisms affecting these metabolite levels.
While the current results highlight important trends, we are committed to expanding our research by incorporating additional variables and larger cohorts. This approach will enhance our understanding of the roles of tryptamine and kynurenine in ASD and their potential as clinically relevant biomarkers.
Reviewer 2 Report
Comments and Suggestions for Authors
The study is very well designed but the field of interest is very specialized.
Autism spectrum disorder (ASD) is a pathology that involves the entire family nucleus and with a significant care commitment and having an additional diagnostic tool could be very useful.
The severity of the patients is described in the study but any other comorbidities or any other antibiotic therapies are not described. The authors are asked to integrate this part.
Furthermore, the clinical relevance appears limited even if the analysis is very well conducted especially for the statistical part.
The study even if with a low number of cases is very interesting but the field of interest is very narrow and specialized.
I would also ask the authors to clarify, if possible according to their experience, whether this diagnostic tool could also allow an earlier diagnosis and therefore a quicker start of treatment.
Very important for the validation of the tool could be to think of a multicentric study to have results with a greater statistical weight.
Author Response
1.The study is very well designed but the field of interest is very specialized.
We acknowledge that our study focuses on a specialized field, which is characteristic of research aimed at developing specific diagnostic tools or biomarkers. The importance of such focused studies is in addressing critical gaps in diagnostic and therapeutic approaches for complex conditions like ASD.
- Autism spectrum disorder (ASD) is a pathology that involves the entire family nucleus and with a significant care commitment and having an additional diagnostic tool could be very useful.
We appreciate the reviewer’s emphasis on the significant impact of ASD on the entire family. This condition often requires intensive caregiving and imposes emotional and financial burdens on families.
The goal of our research is to contribute to the development of diagnostic tools that could enable earlier and more precise diagnosis of ASD. Early diagnosis is essential for starting interventions on time, which can greatly enhance developmental outcomes and alleviate the burden of care on families.
Urinary metabolite ratios linked to the severity of ASD may be found to be a useful adjunct to existing diagnostic techniques. This approach not only enhances diagnostic accuracy but could also provide insights into the unique symptom profiles of individual patients, facilitating more personalized treatment strategies.
A reliable diagnostic tool could significantly reduce the long-term caregiving challenges faced by families by allowing for earlier, more targeted interventions. This would not only improve the child’s developmental trajectory but also enhance the overall quality of life for both the child and their caregivers.We remain committed to advancing research that addresses the challenges faced by families affected by ASD. In our view, our findings represent a meaningful step forward in the development of tools that can improve both the diagnosis and treatment of ASD, ultimately benefiting children with ASD and their families.
- The severity of the patients is described in the study but any other comorbidities or any other antibiotic therapies are not described. The authors are asked to integrate this part.
We appreciate the reviewer’s valuable observation regarding the potential impact of comorbidities and antibiotic therapies on metabolic profiles in children with ASD. These factors are indeed important considerations in biomarker research. While we documented recent antibiotic use and significant comorbidities in our sample, these factors were excluded from the current analysis to maintain the focus on the primary research question. However, we acknowledge the importance of these factors and agree that they warrant further exploration in future studies.
Data on comorbid illnesses, like gastrointestinal disorders, which are prevalent in children with ASD, might be included to shed light on how these factors interact with tryptophan metabolism. Likewise, the use of antibiotics, which affect gut flora, may have a major impact on urine metabolite profiles and needs more research.
The goal of future research will be to gather and examine comprehensive data on antibiotic treatments and comorbidities. We aim to gain a better understanding of how these factors affect metabolic profiles in ASD by working with multidisciplinary teams and performing research with bigger sample numbers.
In order to enhance the depth and therapeutic relevance of our findings, we are committed to addressing these limitations in future research.. Research on metabolic biomarkers in ASD could be significantly advanced by a more thorough understandingof the role of comorbidities and antibiotic use.
The following paragraph has been added to the text:
To ensure a comprehensive analysis, we documented recent antibiotic use and significant comorbidities within our study sample. However, these factors were excluded from the current analysis to maintain the focus on broader metabolic trends. Comorbid conditions, such as gastrointestinal disorders prevalent among children with ASD, and the potential impact of antibiotics on gut microbiota and subsequent metabolic profiles are recognized as critical variables in biomarker research.
Future investigations will aim to integrate these factors by gathering detailed data on antibiotic treatments and comorbidities. This approach will enable a more nuanced understanding of how such variables interact with tryptophan metabolism and influence urinary metabolite profiles. By collaborating with multidisciplinary teams and expanding the sample size, we aim to strengthen the robustness and clinical relevance of our findings.
- Furthermore, the clinical relevance appears limited even if the analysis is very well conducted especially for the statistical part.
We recognize that our findings may initially appear to have limited clinical applicability. However, this is a characteristic feature of exploratory research aimed at identifying biomarkers, where rigorous statistical analysis provides the foundation for subsequent translational investigations. As a foundation for discovering new biomarkers that could help with diagnosis and categorization of ASD severity, our work identifies changes in tryptophan metabolism and related pathways that correlate with the severity of ASD.
With further validation, the identified metabolomic ratios could be developed into supplemental diagnostic tools, enabling earlier detection and more personalized treatment strategies for children with ASD.
To ascertain if these biomarkers might predict the development or progression of ASD, we ecommend implementing longitudinal designs and carrying out bigger, multicentric investigations to validate our findings across diverse populations and provide greater statistical robustness.We remain optimistic that our research represents a significant step toward the development of diagnostic tools that could improve the lives of individuals with ASD and their families. While further studies are required to confirm and expand upon our results, we believe this work lays important groundwork for advancing ASD diagnosis and care.
- The study even if with a low number of cases is very interesting but the field of interest is very narrow and specialized.
We acknowledge that the study focuses on a specific and limited area. However, we emphasize the importance of such targeted research in advancing our understanding of ASD and its underlying mechanisms. By exploring narrow but critical aspects of this complex condition, we aim to contribute to the development of improved diagnostic tools and therapeutic strategies.
- I would also ask the authors to clarify, if possible according to their experience, whether this diagnostic tool could also allow an earlier diagnosis and therefore a quicker start of treatment.
The primary aim of our study is to develop a diagnostic tool that facilitates earlier detection of ASD, thereby enabling timely initiation of treatment. Through this study, we seek to deepen our understanding of tryptophan metabolism and the role of its metabolites in ASD. Building on our previous research, which identified associations between heavy metals, porphyrin fractions, oxidative stress markers, and intracranial fluid spaces, we are continuing to investigate metabolic pathways that could serve as early diagnostic markers. This approach has the potential to streamline interventions and improve outcomes for children with ASD.
- Very important for the validation of the tool could be to think of a multicentric study to have results with a greater statistical weight.
We fully agree that multicentric studies are essential for validating our findings across larger and more diverse populations. Collaborative research involving multiple centers will enhance the statistical robustness of our results and ensure their broader applicability. We are currently exploring opportunities for future cooperative research projects aimed at validating the proposed diagnostic tool in various populations and clinical settings. Such studies are a vital step toward establishing reliable biomarkers for ASD.